# ATP Purinergic Receptor P2X1-Dependent Suicidal NETosis Induced by *Cryptosporidium parvum* under Physioxia Conditions

**DOI:** 10.3390/biology11030442

**Published:** 2022-03-14

**Authors:** Seyed Sajjad Hasheminasab, Iván Conejeros, Zahady D. Velásquez, Tilman Borggrefe, Ulrich Gärtner, Faustin Kamena, Anja Taubert, Carlos Hermosilla

**Affiliations:** 1Institute of Parasitology, Biomedical Research Center Seltersberg (BFS), Justus Liebig University Giessen, 35392 Giessen, Germany; ivan.conejeros@vetmed.uni-giessen.de (I.C.); zahady.velasquez@vetmed.uni-giessen.de (Z.D.V.); anja.taubert@vetmed.uni-giessen.de (A.T.); carlos.r.hermosilla@vetmed.uni-giessen.de (C.H.); 2Institute of Biochemistry, Justus Liebig University Giessen, 35392 Giessen, Germany; tilman.borggrefe@biochemie.med.uni-giessen.de; 3Institute of Anatomy and Cell Biology, Justus Liebig University Giessen, 35392 Giessen, Germany; ulrich.gaertner@anatomie.med.uni-giessen.de; 4Laboratory for Molecular Parasitology, Department of Microbiology and Parasitology, University of Buea, Buea P.O. Box 63, Cameroon; faustinkamena@googlemail.com

**Keywords:** *Cryptosporidium parvum*, NETosis, purinergic receptor P2X1, Notch, glycolysis, PMN

## Abstract

**Simple Summary:**

*Cryptosporidium parvum*, a zoonotic apicomplexan, is one of the leading causes of severe diarrheal disease worldwide and a major contributor to early infant and neonatal mortality. In vivo, sporozoites of *C. parvum* must invade enterocytes where they develop in a unique intracellular extracytoplasmic niche. Polymorphonuclear neutrophils (PMN) are the most common leukocytes in the immune defense system of the host, the first to be transferred to *C. parvum*-infected gut mucosa, and are increasingly recognized as important cells in the fight against other intestinal parasites. We investigated the role of ATP purinergic receptor P2X1, glycolysis, plasma membrane monocarboxylate transporters (MCTs) of lactate, extracellular acidification rates (ECAR), and mitochondrial oxygen consumption rates (OCR), as well as Notch signaling in *C. parvum*-triggered NETs formation in exposed bovine PMN under intestinal physioxic (5% O_2_) as well as hyperoxic (21% O_2_) conditions. Both *C. parvum* stages, i.e., sporozoites and oocysts, strongly induce suicidal NETosis in a P2X1-dependent manner, suggesting anti-cryptosporidial effects not only through firm sporozoite entrapment and hampered sporozoite excystation, but also probably through direct exposure to NETs-derived toxic molecules.

**Abstract:**

Cryptosporidiosis is a zoonotic intestinal disease that affects humans, wildlife, and neonatal cattle, caused by *Cryptosporidium parvum.* Neutrophil extracellular traps (NETs), also known as suicidal NETosis, are a powerful and ancient innate effector mechanism by which polymorphonuclear neutrophils (PMN) battle parasitic organisms like protozoa and helminths. Here, *C. parvum* oocysts and live sporozoites were utilized to examine suicidal NETosis in exposed bovine PMN under both 5% O_2_ (physiological conditions within small intestinal tract) and 21% O_2_ (normal hyperoxic conditions in research facilities). Both sporozoites and oocysts induced suicidal NETosis in exposed PMN under physioxia (5% O_2_) and hyperoxia (21% O_2_). Besides, *C. parvum*-induced suicidal NETosis was affirmed by total break of PMN, co-localization of extracellular DNA decorated with pan-histones (H1A, H2A/H2B, H3, H4) and neutrophil elastase (NE) by means of confocal- and immunofluorescence microscopy investigations. *C. parvum*-triggered NETs entrapped sporozoites and impeded sporozoite egress from oocysts covered by released NETs, according to scanning electron microscopy (SEM) examination. Live cell 3D-holotomographic microscopy analysis visualized early parasite-induced PMN morphological changes, such as the formation of membrane protrusions towards *C. parvum* while undergoing NETosis. Significant reduction of *C. parvum*-induced suicidal NETosis was measured after PMN treatments with purinergic receptor P2X1 inhibitor NF449, under both oxygen circumstances, this receptor was found to play a critical role in the induction of NETs, indicating its importance. Similarly, inhibition of PMN glycolysis via 2-deoxy glucose treatments resulted in a reduction of *C. parvum*-triggered suicidal NETosis but not significantly. Extracellular acidification rates (ECAR) and oxygen consumption rates (OCR) were not increased in *C. parvum*-exposed cells, according to measurements of PMN energetic state. Treatments with inhibitors of plasma membrane monocarboxylate transporters (MCTs) of lactate failed to significantly reduce *C. parvum*-mediated NET extrusion. Concerning Notch signaling, no significant reduction was detected after PMN treatments with two specific Notch inhibitors, i.e., DAPT and compound E. Overall, we here describe for the first time the pivotal role of ATP purinergic receptor P2X1 in *C. parvum*-mediated suicidal NETosis under physioxia (5% O_2_) and its anti-cryptosporidial properties.

## 1. Introduction

*Cryptosporidium parvum* is a zoonotic apicomplexan parasite that belongs to the family Cryptosporidiidae of the phylum Alveolata (subphylum Apicomplexa) and has a global distribution. *C. parvum* is considered an important enteropathogen, exposing mainly young children (<5 years of age), immunocompromised humans, and neonatal livestock at risk of severe enteritis [1,2,3,4,5,6,7]. According to a study, *C. parvum* is the second most common cause of newborn diarrhoea, which is linked to toddler mortality in impoverished countries [8]. Chronic cryptosporidiosis could become a deadly condition in immune-compromised persons if not treated properly [8,9,10,11,12]. As seen in humans, *C. parvum* infection in neonatal calves results in severe enteritis with high economic losses for the cattle industry worldwide [13,14]. More importantly, cattle represent the most important reservoir host species for *C. parvum,* thus playing a crucial role in the epizootiology of human cryptosporidiosis [13,14,15].

The bulk of immunological investigations into the protective host’s immune responses to *C. parvum* in vivo concentrated on adaptive cell-mediated immune reactions. Nonetheless, most of these data originated from *C. parvum*-infected mouse models, which differ significantly from the immune systems of humans and livestock, and therefore do not properly reflect in vivo immune reactions of humans/livestock as previously demonstrated [2,5,7,16]. Unlike adaptive immunity, less information is available regarding initial host innate immunological responses to *C. parvum*, notably among polymorphonuclear neutrophils (PMN) and other closely related professional phagocytes [2]. Several findings on host innate immunity have shown that PMN, monocytes, macrophages, and intestinal epithelial cells (IEC) play a key role in protection against this parasite [2,5,9,10,17,18,19,20,21,22,23,24]. PMN are the most prevalent leukocyte population in the mammalian bloodstream and serve as the first defense line. Consistently, PMN are the first ones to be recruited to sites of *C. parvum* infections [25,26], as also reported for closely related ruminant intestinal apicomplexans (i.e., *Eimeria bovis*, *Eimeria arloingi*) [2,7]. PMN constitutively express pathogen recognition receptors (PRR), including various Toll-like receptors (TLR) [2,27,28], dectin-1 [27], and CD-11b on their surfaces [29,30], as well as cytosolic PRR-recognizing pathogen-associated molecular patterns (PAMP). Furthermore, circulating PMN express the retinoid acid-inducible gene-I (RIG-I)-like ligands (RLR) and NOD as well [31,32]. Thus, PMN can swiftly detect PAMP, which include microbial membrane components such as lipoproteins (TLR2 ligands), lipopolysaccharide (LPS; TLR4 ligand), flagellin, bacterial nucleic acids, fungal, and parasitic-derived molecules [32,33]. Following PAMP-mediated identification, PMN-derived effector processes will quickly start, such as degranulation [34], the generation of reactive oxygen species (ROS) [6,27,28,35], phagocytosis [36], and neutrophil extracellular traps (NETs) that are extruded by neutrophils to entrap and eliminate invading microorganisms [25,26,37,38]. Extruded NETs are comprised primarily of nuclear DNA decorated with global histones (H1-4), various anti-bacterial granular effector molecules (e.g., lactoferrin, pentraxin, cathelisin, and gelatinase), and are extruded through a unique cell death process called suicidal NETosis, which is mediated by intrinsic NADPH oxidase [17].

Referring to parasites, suicidal NETosis has been the most observed phenotype of NETs against multiple intracellular protozoans (e.g., *Eimeria bovis*, *Toxoplasma gondii*, *Neospora caninum*, *Besnoitia besnoiti*, *Leishmania amazonensis*) as well as extracellular protozoans (*Trypanosoma cruzi*, *Trypanosoma brucei*) [2,7,28,37,38,39,40,41,42,43,44,45], large nematodes (e.g., *Haemonchus contortus*, *Strongyloides stercoralis*, *Angiostrongylus vasorum*, *Aelurostrongylus abstrusus*, *Troglostrongylus brevior*, *Brugia malayi*, *Dirofilaria immitis*) and trematodes, (e.g., *Schistosoma japonicum*, *Fasciola hepatica*) [6,46]. Additionally, protozoan and helminth-induced NETs resulted in a variety of NET morphologies, including spread NETs (sprNETs), diffuse NETs (diffNETs), aggregated NETs (aggNETs), and cell free- and anchored NETs [6,28,40,44]. Despite numerous publications on parasite-induced NETs, there has only been one report on *C. parvum*-induced suicidal NETosis in humans and cattle [47]. Accordingly, *C. parvum*-mediated NETs were shown to firmly entrap highly motile sporozoites in vitro and, furthermore, to cover oocysts, thereby hampering sporozoite excystation. More notably, *C. parvum* sporozoites treated with NETs had much lower in vitro infectivity for IEC [47]. The upregulation of GM-CSF, IL6, CXCL8, and TNF-α gene transcription in response to *C. parvum*-sporozoite stimulation, as well as the importance of p38 MAPK, NOX, NE, MPO, Ca++ influx and ERK1/2 in *C. parvum*-induced NETosis, has been documented [47]. Nonetheless, former *C. parvum*-related NETosis experiments were performed under hyperoxia (21% O_2_), which does not reflect physiological oxygen concentrations (physioxia; 1–11% O_2_) of the small intestine in vivo [48,49]. Consequently, as microaerobic intestinal *C. parvum* replication occurs under physioxia, this fact should be considered to better understand not only the metabolic requirements of *C. parvum* [4,12], but also the impact of physioxia (5% O_2_) on PMN-, monocyte-, macrophage- and IEC-derived effector mechanisms as previously suggested [4,12].

Notch signaling is well-known for its role in numerous conserved signal transduction pathways of immune cell development and homeostasis, and it has been connected to many features of peripheral T-cell immunological reactions and T-cell differentiation [50,51]. Similarly, monocytes, macrophages, and dendritic cells (DCs) produce Notch ligands and receptors constitutively, allowing them to trigger and react to Notch signals by TLR regulation [52,53]. To better understand Notch functions in PMN, we here investigated Notch signaling in *C. parvum*-triggered NETosis by using specific inhibitors and further assessed the impact of oxygen conditions on PMN-derived glycolysis, ATP purinergic P2X1 receptor and monocarboxylate transporters (MCTs). Finally, the visualization of dynamic *C. parvum*-induced suicidal NETosis was conducted by live cell imaging 3D-holotomography analysis under physioxia (5% O_2_) to be as close as possible to the in vivo intestinal scenario.

## 2. Materials and Methods

### 2.1. Ethics Statements

This study was carried out in accordance with the guidelines of the Justus Liebig University (JLU) Giessen Animal Care Committee. Protocols were authorized by the Federal State of Hesse’s Ethic Commission for Experimental Animal Studies (Regierungspräsidium Giessen; A9/2012; JLU-No.521 AZ), Germany, and in compliance with European Animal Welfare Legislation: ART13TFEU and presently relevant German Animal Protection Laws.

### 2.2. Cryptosporidium parvum Strain and Sporozoite Excystation

Serial passages of oocysts in one-day-old calves at the Institute of Parasitology, University of Leipzig, Leipzig, Germany, were used to maintain the *C. parvum* strain (IIaA15G2RI), as previously described [46]. This strain belongs to the 60-KDa glycoprotein (gp60) IIaA15G2RI subtype of *C. parvum*, which is the most often described zoonotic *C. parvum* subtype in Germany and other industrialized nations [4,12,54,55,56]. *C. parvum* oocysts were isolated using a combination sedimentation/flotation approach as outlined previously [4,57], and then kept at 4 °C in sterile phosphate-buffered saline (PBS; pH 7.4) mixed with penicillin/streptomycin (200 μg/mL; Sigma-Aldrich, Darmstadt, Germany) and amphotericin B (5 μg/mL; Sigma-Aldrich, Darmstadt, Germany). As previously reported, the medium of *C. parvum* oocyst stock was replaced at monthly intervals [4,12].

To isolate sporozoites, *C. parvum* oocysts were pelleted at 5000 g for 5 min at 4 °C before being suspended in excystation medium [4,57]. In summary, acidified (pH 2.0) and sterile pre-warmed (37 °C) 1x Hank’s balanced salt solution (HBSS, Sigma-Aldrich, Darmstadt, Germany) supplementation for 10–30 min at 37 °C produced *C. parvum* sporozoite excystation. Following that, free-released sporozoites were pelleted (5000 g for 5 min) and incubated for 10 min at 37 °C in non-acidified 1x HBSS (Sigma-Aldrich, Darmstadt, Germany). After a final centrifugation (5000× *g* for 5 min), sporozoites were resuspended in sterile RPMI 1640 cell culture media (Gibco, Darmstadt, Germany) without phenol red and treated with 0.3 g/mL L-glutamine, 10% foetal bovine serum (FBS; both Gibco, Darmstadt, Germany), 100 UI penicillin, and 0.1 mg streptomycin/mL (both Sigma-Aldrich, Darmstadt, Germany). For bovine PMN exposure and NETosis-related experiments, newly and motile sporozoites were employed.

### 2.3. Bovine PMN Isolation

Adult healthy dairy cattle (*n* = 9) from the Oberer Hardthof research animal farm, JLU Giessen, Germany, acted as blood donors. Jugular vein puncture was used to acquire peripheral blood (30 mL in 12 mL heparinized sterilized plastic tubes) (KABE Labortechnik, Nümbrecht, Germany). A second step included diluting 20 mL heparinized blood with 0.02 percent ethylenediaminetetraacetic acid (EDTA; Sigma-Aldrich, Darmstadt, Germany), layering it on top of 12 mL Biocoll^®^ separation solution (density = 1.077 g/L; Biochrom AG, Berlin, Germany), and centrifuging it for 45 min at 800× *g*. Cells were re-suspended in 25 mL phosphate buffer lysis and gently stirred for 40 s to lyse erythrocytes after the removal of plasma and peripheral blood mononuclear cells (PBMC). Adding 4 mL of sterile 10x HBSS (Sigma-Aldrich, Darmstadt, Germany) quickly restored osmolarity. Following an additional round of erythrolysis, re-suspension was repeated in sterile 10x HBSS. As previously reported, PMN were counted in a Neubauer haemocytometer and kept 30 min at 37 °C in a 5% CO_2_ environment before being used in the experiments [58].

### 2.4. The Determination of Oxygen Consumption Rates (OCR) and Extracellular Acidification Rates (ECAR) in Bovine PMN That had Been Exposed to Cryptosporidium parvum

After being exposed to sporozoites, the energetic state of PMN was studied by a Seahorse XF^®^ analyzer (Agilent, Rathingen, Germany). At the beginning, 1 × 10^6^ PMN from three animals (*n* = 3) were pelleted for 10 min at 500× *g* (RT). Resuspension of pellet was performed in 0.5 mL XF^®^ assay media (Agilent, Rathingen, Germany) supplied with, 1 mM pyruvate, 10 mM glucose, and 2 mM L-glutamine after supernatant removal (all Agilent, Rathingen, Germany). In total, 1 × 10^5^ cells (50 µL of solution), were gently pipetted in each well of an eight-well XF^®^ Seahorse analyser plate (Agilent, Rathingen, Germany), that had been pre-coated with 0.001% poly-L-lysine for 30 min (Sigma-Aldrich, Darmstadt, Germany). Thereafter, 50 μL of XF^®^ assay medium (Agilent, Rathingen, Germany) was applied to the empty wells and plain XF^®^ assay medium (20 mL, Agilent, Rathingen, Germany) was used as a control for PMN in the study. At the end, to reach the final 180 µL per well, 130 µL of XF^®^ assay media (Agilent, Rathingen, Germany) was added to each well, following incubation at 37 °C with absence of CO_2_ supplementation for 45 min before being measured using the Seahorse XF^®^ assay. Moreover, *C. parvum* (3 × 10^5^ oocysts/20 μL) were suspended in XF^®^ assay medium (Agilent, Rathingen, Germany) and inserted in one of the injection ports to allow parasite exposure to bovine PMN. As previously mentioned, the area under the curve (AUC) and the evaluation of OCR, ECAR of collected registers were carried out using Wave^®^ software (Desktop Version, Agilent, Rathingen, Germany) [6,40].

### 2.5. Inhibition of ATP Purinergic Receptor P2X1, MCT1, MCT2, and Glycolysis in Cryptosporidium parvum-Exposed Bovine PMN

Bovine PMN were pre-treated with the mentioned inhibitors for 30 min and then co-cultured with *C. parvum* sporozoites (1:3 PMN/sporozoite ratio, 3 h, 37 °C) with the purpose of blocking the ATP purinergic receptor P2X1, MCT1, MCT2, and glycolysis under hyperoxia 21% O_2_ = oxygen conditions widely used in most *C. parvum*-related studies) and physioxia (5% O_2_) conditions, simulating physiological oxygen pressure of small intestine in vivo as follows [4,12,54,59]: AR-C 155858 (1 μM, MCT1- and MCT2-inhibitors, Tocris Bioscience, Wiesbaden, Germany), AR-C 141990 (1 μM, MCT1 inhibitor, Tocris Bioscience, Wiesbaden, Germany), NF449 (100 μM, purinergic receptor P2X1 antagonist, Tocris Bioscience, Wiesbaden, Germany), oligomycin A (5 μM, inhibitor of mitochondrial respiration, Sigma-Aldrich, Darmstadt, Germany) and 2-DG (2 mM; antagonist of glycolysis, Sigma-Aldrich, Darmstadt, Germany), were used in this study [37,44,60,61].

### 2.6. Inhibition of Notch Signaling in Cryptosporidium parvum-Exposed Bovine PMN under Physioxia and Hyperoxia

These studies evaluated the impact of Notch signaling inhibitors on *C. parvum*-mediated suicidal NETosis. Therefore, PMN and *C. parvum* sporozoites (1:4) were co-cultured in 5%O_2_ (physioxia) [4,55,56] and 21%O_2_ (hyperoxia) conditions [4,12]. As previously reported by Vélez et al. [4,12], in vitro physioxic culture systems were achieved in an InvivO_2_^®^ 400 physiological cell culture workflow unit (Ruskinn, Vienna, Austria), whereas hyperoxic (21%O_2_) co-culture situations were achieved through standardized cell culture incubator (Heracell 240i, Thermo Scientific, Langenselbold, Germany). For the purposes of this study, 1 × 10^6^ cells/mL (*n* = 3) were suspended in sterile 1x HBSS buffer (Sigma-Aldrich, Darmstadt, Germany) under both oxygen conditions. It was necessary to add SYTOX Green^®^ (5 µM; Thermo Fisher Scientific, Braunschweig, Germany), and cells were seeded (10^5^ cells in 50 μL/each well) in a microplate (96-well plate; Greiner, Frickenhausen, Germany). The inhibitors of Notch signaling compound E and DAPT (Merck Millipore, Darmstadt, Germany) were added (for DAPT, 50 μM and for compound E, 20 nM). *C. parvum* sporozoites (3 × 10^5^ sporozoites/well) were cultured under physioxia and hyperoxia, pelleted, and re-suspended in the same buffer as described earlier. This study used an automated multiplate monochrome reader (Varioskan Flash^®^; Thermo Scientific, Langenselbold, Germany) to track the formation of NETs over a period of 120 min with 504 nm excitation and 523 nm emission wavelength. The results were recorded for 120 min every 2 min.

### 2.7. Examination of Cryptosporidium parvum-Induced NETs Using Scanning Electron Microscopy (SEM) Analysis

Viable *C. parvum* oocysts/sporozoites (proportion 1:3) were co-cultured with bovine PMN for 120 min on pre-coated coverslips with poly-L-lysine 0.01% (10 mm/diameter; Thermo Fisher Scientific, Braunschweig, Germany). The co-cultures were performed at 37 °C and 5% CO_2_ for the entire period. The cells were fixed in 2.5% glutaraldehyde (Merck, Darmstadt, Germany) and also post-fixed with 1 percent osmium tetroxide (Merck, Darmstadt, Germany), rinsed in distilled water, dried, CO_2_-treated to the critical point, and lastly sputtered with gold particles. The Institute of Anatomy and Cell Biology at the JLU Giessen utilized a scanning electron microscope (Philips XL30^®^, Eindhoven, The Netherlands) to investigate gold-sputtered materials.

### 2.8. Cryptosporidium parvum-Induced Suicidal NETosis Visualization Using Immunofluorescence- and Confocal Microscopy Analyses

Bovine PMN and *C. parvum* sporozoites and oocysts (proportion 1:3) were co-cultured (37 °C and 5% CO_2_, 21% and 5% O_2_) for 120 min on sterile glass coverslips (15 mm-diameter, Thermo Fisher Scientific, Braunschweig, Germany) that had been pre-treated with 0.01% poly-L-lysine (Sigma-Aldrich; Darmstadt; Germany). The cell fixation was conducted in 4% paraformaldehyde (Merck, Darmstadt, Germany) and kept at 4 °C and then suicidal NETosis was visualized using DAPI (Thermo Fisher Scientific, Braunschweig, Germany). To identify NETs-specific components and proteins, anti-neutrophil elastase (NE) antibodies (1:500; Abcam-ab68672, Merck-MAB3422, Darmstadt, Germany), and global anti-histone antibodies (1:500; Darmstadt, Germany) were employed. This was followed by three sterile PBS (Sigma-Aldrich, Darmstadt, Germany) washes, a 60-min incubation time in bovine serum albumin (2% BSA, (Sigma-Aldrich, Darmstadt, Germany)) supplemented with 0.3% Triton X100 (Thermo Fisher Scientific, Braunschweig, Germany), and then incubation with primary antibodies overnight. The coverslips were again incubated in secondary antibody solutions (1:500: IgG #A11005-Alexa 594 goat anti-mouse and IgG #A110011-Alexa 488 goat anti-mouse, Thermo Fisher Scientific, Braunschweig, Germany) for 60 min at room temperature (RT) and in total darkness after three washing steps with sterile PBS (Sigma-Aldrich, Darmstadt, Germany). Anti-fading buffer with DAPI (Thermo Fisher Scientific, Braunschweig, Germany) was used to mount the samples after they had been rinsed three times with sterile PBS. A confocal microscope (Nikon ECLIPSE Ti2, Melville, NY, USA) and an inverted epifluorescence microscope IX81^®^ (Olympus, Tokyo, Japan) integrated with a XM10^®^ digital camera (Olympus, Tokyo, Japan) were both used to observe *C. parvum*-mediated suicidal NETosis in the experiments.

### 2.9. 3D Holotomographic Microscopy Investigation of Cryptosporidium parvum-Induced NETosis in Live Cells

For this experiment, we pelleted 1 × 10^6^ of bovine PMN through 300× *g* centrifugation/10 min at RT. After removing the supernatant from the cell pellets, the cells were re-suspended in an imaging media that included 0.5 μM SYTOX Green^®^ (Thermo Fisher Scientific, Braunschweig, Germany), 2 μM DRAQ5 (Thermo Fisher Scientific, Braunschweig, Germany), and 0.1% BSA (Sigma-Aldrich, Darmstadt, Germany). Sterilized ibidi^®^ plastic cell culture plates (diameter 35 mm^2^ with low profile, ibidi, Gräfelfing, Germany) were used to seed one ml of this cell solution, which was incubated at 5% CO_2_ and 37 °C in an incubation cell chamber (ibidi^®^ -Dish 35 mm). For PMN settling and to prevent condensation on the chamber lid, a 30-min resting period was employed. Sporozoites from *C. parvum* (1.5 × 10^6^) were introduced to the cell culture. Using a Fluo-3D Cell Explorer^®^ (Nanolive, Lausanne, Switzerland) image capture was configured for refractive index (RI), DRAQ5^®^ (red channel) and SYTOX Green^®^ (green channel) detection (Thermo Fisher Scientific, Braunschweig, Germany) acquiring images for 3 h every minute. At the conclusion of the experiment, each channel was exported independently and processed using Image J^®^ software using Steve^®^ software v.2.6 (Nanolive, Lausanne, Switzerland) (Fiji version 1.7; NIH). Using the RI values from the photos, digital staining was also accomplished [4].

### 2.10. Statistical Methods

Statistical significance was defined by a *p*-value of <0.05 and the following analysis was performed by applying: parameters were assessed and Kruskal–Wallis test followed by Dunn’s post-hoc test for multiple comparisons. GraphPad^®^ Prism software (v.9.1), was used for all graphs (mean ± SD), AUC calculations, and statistical analyses (v.7.03).

## 3. Results

### 3.1. Cryptosporidium parvum-Oocysts and Sporozoites Induced Suicidal NETosis

For the ultrastructural detection of *C. parvum*-triggered NETosis, SEM analyses were performed. Suicidal NETosis was confirmed by SEM analysis, which revealed that exposure of bovine PMN to *C. parvum* oocysts (Figure 1A–D) for 120 min triggered the development of both thick and thin fibers of chromatin strands being released from PMN (Figure 1A–D, blue arrows). SEM analysis also demonstrated that not all exposed PMN responded by NET formation as some remained non-activated showing smooth PMN surfaces (Figure 1B, white arrow head) whereas others seeming to be activated thereby showing rough/irregular surfaces (Figure 1B,D, indicated by black arrow heads). Some *C. parvum*-oocysts seemed to be loosely covered by extruded NETs fibers (Figure 1A), but others were almost completely coated by NETs (Figure 1D, red arrow). The same was true for the sporozoites caught in PMN-derived extracellular fibers, demonstrating that *C. parvum*-mediated NET production is not a stage-specific defense mechanism and verifying that NETs may hamper both sporozoite excystation and sporozoite host cell invasion, as previously shown [2].

Immunofluorescence and confocal microscopy studies using antibodies against global histones and NE, as well as DNA staining, validated the traditional hallmarks of mammalian-derived NETs in stimulated bovine PMN induced by *C. parvum*. DNA-positive extracellular fiber backbones co-localized with H1-, H2A/B-, H3-, H4-, and NE-positive signals, thereby proving *C. parvum*-triggered NETs (Figure 2). Confocal microscopy, as well as 3D holotomographic imaging of bovine PMN during NET formation, validated these findings (Figure 3 and Appendix A).

### 3.2. Live Cell 3D-Holotomography Illustrated Cryptosporidium parvum-Mediated NETosis

Additionally, we used live cell three-dimensional holotomographic microscopy (3D Cell Explorer^®^, Nanolive, Lausanne, Switzerland) to see and better comprehend the early contacts between bovine PMN and *C. parvum* oocysts/sporozoites throughout the dynamic NETotic process. After the first 30 min of exposure with either motile sporozoites or static oocysts, bovine PMN activated, revealing pseudopod/laminopod production and fast movement (crawling) actions of PMN towards power vision regions containing parasite stages. For exemplary illustrations, interactions of bovine PMN with parasites, including the morphology of non-activated bovine PMN, are depicted in Figure 4, all being complemented with live cell 3D rendering and RI-based digital staining. Thus, some conventional PMN forms, including a polysegmented nucleus, may be differentiated (red). Imaging was further complemented by live cell imaging 3D-holotomographic footage conducted under hyperoxia (21% O_2_), and Video 3 contains the whole footage of this experiment. This signal was highly linked with a change in the nuclear structure of PMN as determined by DRAQ5^®^ labeling (red). To supplement previously acquired findings from immunofluorescence microscopy analysis of fixed cells, then a live cell imaging 3D-holotomography research was carried out to obtain better understanding of parasite-PMN contacts and triggered morphological nuclear alterations. The time-lapse experiment is depicted in Figure 4. After 120 min of co-culture, an increment in SYTOX Green^®^ signals (green) was detected. In the case of DRAQ5^®^ staining, this signal was associated with a shift in nuclear form (red). Experiments involving PMN exposed to *C. parvum* revealed an increase in cytoplasmic size and a shift in nuclear structure.

### 3.3. Sporozoite Exposure had No Effects on Extracellular Acidification Rates (ECAR), Glycolysis, or Oxygen Consumption Rates (OCR) in Bovine PMN

We conducted a series of tests to assess the metabolic activities of *C. parvum*-exposed bovine PMN (*n* = 3) by the use of an extracellular flux analyzer Seahorse XFp^®^ (Agilent, Rathingen, Germany), which assesses OCR and ECAR rates in vital cells. As a result, both OCR and ECAR were computed using cell energy profiles. In this experiment, calcium ionophore A23187 was used as a positive control (Figure 5). As indicated in Figure 5, after baseline OCR and ECAR values for plain PMN were obtained, the addition of live parasites did not result in a rapid and persistent rise in ECAR values. When estimating the AUC, neither enhanced ECAR- nor OCR values were detected in *C. parvum*-exposed PMN when compared to non-stimulated PMN. This assumption was underlined by the fact that pre-treatments of PMN with oligomycin (an inhibitor of ATP synthase within mitochondrial respiration) failed to affect *C. parvum*-mediated NETs formation (see Figure 6).

To further verify the importance of glycolysis on the NETotic process, we additionally pre-treated bovine PMN with 2-DG, an inhibitor of glycolysis, and afterwards exposed them to parasite stages. In line with the above-mentioned data, these treatments did not reduce *C. parvum*-triggered NETs formation (treated vs. non-treated condition), suggesting this process as a glycolysis-independent innate effector mechanism (Figure 7).

### 3.4. Cryptosporidium parvum-Induced Bovine NETosis Is a MCT1- and MCT2-Independent Cell Death Process

Given that lactate is a major metabolite of glycolysis and may contribute to extracellular acidification after being effluxed via MCTs found in the plasma membrane of bovine PMN [44]. Additionally, we investigated the involvement of MCT1/MCT2 in *C. parvum*-mediated NETosis. Because of this, PMN were treated with inhibitors of MCT1 (AR-C 155858) or MCT1/2 (AR-C 141990) prior to parasite exposure in order to perform inhibition experiments. Even though both inhibitors resulted in a diminishment of NETosis under 21% O_2_ conditions (Figure 6), no statistically significant differences were seen between non-treated controls and ARC-155858-treated PMN, or between non-treated controls and ARC-14190-treated PMN (*p* > 0.05 for both comparisons). When applying 5% O_2_, even fewer effects were stated (Figure 6), thereby implying that these transporters were probably not relevant in *C. parvum*-triggered NETosis under physioxia (5% O_2_) and hyperoxia (21% O_2_), respectively.

### 3.5. Cryptosporidium parvum-Induced NETosis Depends on P2X1-Mediated Purinergic Signaling but Not on Notch-Regulated Signal Transduction

In this research, we investigated the role of P2X1-dependent purinergic signaling in the formation of *C. parvum*-induced suicidal NETosis. PMN were thus pre-treated with NF449, a selective inhibitor of the purinergic receptor P2X1, which prevents ATP-dependent cell signaling, before being added to the culture. It was discovered that NF449 pre-treatments inhibits (Figure 7) the production of *C. parvum*-induced NETs when compared with untreated controls (non-treated PMN + *C. parvum* versus treated PMN + *C. parvum*: *p* < 0.05). According to these findings, purinergic P2X1-mediated ATP binding, and most likely the ensuing cellular signaling are critical for *C. parvum*-induced NETosis.

Notch-mediated cell signaling represents a well-conserved signal transduction pathway mainly being involved in cell differentiation and homeostasis of numerous cell types. So far, the role of Notch in bovine NETosis has not been investigated yet. Given that different Notch inhibitors are preferably used in different host systems [62], we here applied two of the most commonly used inhibitors, i.e., compound E and DAPT, which block the activating proteolytic processing of the receptor (gamma-secretase inhibitors block the cleavage of the N-receptor). Both inhibitors have been successfully used in the murine- and the human- but never in the bovine immune system. After pre-treatment of bovine PMN with both Notch-specific inhibitors, NETs release was quantified after parasite exposure. Overall, both inhibitors led to a reduction of parasite-triggered suicidal NETosis. However, the most prominent effects were induced by compound E (5% O_2_: treated PMN + *C. parvum* vs. non-treated PMN + *C. parvum*: *p* = 0.001; 21% O_2_: treated PMN + *C. parvum* versus non-treated PMN + *C. parvum*: *p* = 0.006) (Figure 8), whereas Notch treatments failed to reach significance (Figure 8A). Of note, both inhibitors seemed more effective under physioxia (Figure 8B). Overall, these findings indicate a pivotal role for oxygen concentration in Notch-dependent signaling in *C. parvum*-induced NETosis.

## 4. Discussion

Meanwhile, numerous reports have studied the crucial role of NETosis against invasive protozoan [2,32,43,44,58,63,64,65,66] and helminth parasites [2,6,40,67,68]. However, so far, there is only one report on *C. parvum*-induced suicidal NETosis in the literature [2]. We have shown that *C. parvum* sporozoite and oocyst-mediated suicidal NETosis occurs by establishing co-localization of nuclear H1A, H2A/B, H3-4, and NE in DNA-rich NET complexes produced by *C. parvum*. In line, current data also confirmed the capability of extruded NETs to firmly entrap *C. parvum* sporozoites and oocysts, which might impair sporozoite active host cell invasion and sporozoite excystation as described before by Muñoz-Caro et al. [2]. Given that *C. parvum* obligatorily needs to invade host IEC for further intracellular development, it appears feasible to suggest NETosis as an important early host defence mechanism in vivo as being reported for other closely related ruminant intestinal coccidian parasites, i.e., *E. bovis* and *E. arloingi* [7]. Current findings on oocyst-triggered NETosis corresponded well to *E. arloingi* oocyst-triggered NET formation, which resulted in blockage of oocyst micropyles, thereby hampering sporozoite excystation [64]. Consequently, *C. parvum* oocyst covered by NETs might also abrogate sporozoite egress process, thereby blocking the life cycle at an onset of infection as postulated for eimerian parasites [7,64].

To corroborate NETosis findings in living cell experiments, we used a 3D Cell Explorer^®^ microscope (Nanolive, Lausanne, Switzerland) and an upper chamber to maintain constant temperature and CO_2_ atmospheric circumstances. In summary, our method enabled the development of digital stained photos and 3-D renderings of dynamic NETotic processes, including resting PMN, active cells, and PMN in close proximity to *C. parvum* as well as cells forming NETs. Following *C. parvum* infection, typical morphological characteristics of activated PMN were identified, including the extensive presence and mobility of cytoplasmic granules, nuclear alterations, adhesion, rough surface membranes, crawling movements, and laminopod development targeting parasites. In line, *C. parvum* stages with PMN co-cultures also induced the NETotic process, confirming previous reports on this initial nuclear event [6,27,28,40,44,69]. Surprisingly, 3D-digital staining revealed nuclear degeneration 180 min following *C. parvum* exposure and demonstrated a late phase of the NETotic process, similar to *B. besnoiti*-mediated NETosis in exposed bovine PMN [28,44,69].

In vivo, *C. parvum* stages (i.e., sporozoites, trophozoites, merozoites, and gametocytes) reside in a very specialized niche through their epicellular intracellular but extracytoplasmic location, thereby being separated only by a thin membrane from the intestinal contents. There are many commensal and pathogenic microorganisms [70] in the gut that make it a unique ecosystem with its own physiological oxygen pressure atmosphere known as “physioxia/normoxia,” and which is one of the largest human organ surfaces [48,71]. With numerous studies proving the value of physioxic circumstances in the NETs production [72], the metabolism of gut epithelium [73,74], as well as on the metabolic signatures of *C. parvum*-infected primary bovine small intestinal epithelial cells (BSIEC) and ex vivo explants of bovine small intestine (BSI), the concern of intrinsic pathological environment has achieved significant importance in recent years [4,12]. Therefore, we investigated the metabolic signature of *C. parvum*-induced NETosis under both oxygen conditions (physioxia-5% O_2_ and hyperoxia-21% O_2_). PMN in humans and cattle have a variety of fast-acting effector mechanisms including respiratory burst, phagocytosis, NETosis and degranulation, all of which are fueled by high energy and metabolic demands to function properly [37,44,75]. Only three studies (to our knowledge) have been published on the metabolic demand of PMN during parasite-induced NETosis dealing with *N. caninum*- [37], *B. besnoiti*- [44] and *T. b. brucei* [40]. Consequently, *N. caninum*- and *T. b. brucei*-induced NETosis was reported to depend on purinergic receptor-mediated ATP binding [37,40], whilst a dependency on MCT1/2-, P2X1- and glycolysis was stated for *B. besnoiti*-triggered NETs formation [44]. In this investigation, we sought to better understand the role of particular metabolic pathways in parasite-induced suicidal NETosis in intestinal physioxia by mimicking real circumstances in the small intestine [4,12,48,49]. Current data reveals that parasite exposure did not boost glycolytic responses in bovine PMN; however, glycolysis inhibitor treatments (2-DG) did not reduce *C. parvum*-mediated NETs formation. Similar findings were stated for *B. besnoiti* tachyzoites, where FDG treatments reduced NETosis [44]. Of note, increased glycolytic activity is generally linked to enhanced lactate production and release. MCTs are responsible for the transfer of lactate throughout cell membranes, including those of PMN. The MCT family consists of 14 members, with MCT1-4 serving as the primary transporters for lactate absorption and outflow in both humans and animals [76,77,78]. However, inhibition of MCT1 and MCT2 via chemical blockers (ARC-15858 and ARC-141990) had no statistically significant impact on NETosis triggered by *C. parvum*. MCT1 is often involved in the import of lactate in oxidative cells, while MCT4 is more involved in the export of lactate obtained from aerobic glycolysis. As a result, according to Alarcón et al., additional transporters other than MCT1/2 may be implicated in parasite-driven NETosis as well [79]. Nevertheless, by tendency, a reduction of NET formation was apparent after MCT1/2-inhibitor treatments, thereby indicating that lactate efflux may indeed be of importance in order to prevent adverse cytosol acidification due to enhanced glycolytic activities after *C. parvum* stimulation. Interestingly, acidosis due to enhanced lactic acid production was already reported in neonatal calves suffering from acute cryptosporidiosis [80]. These data may indirectly support enhanced glycolytic responses not only in circulating PMN but also in recruited PMN into *C. parvum*-infected gut mucosa.

Besides glycolysis, purinergic receptors for ATP are well-known to be involved in various essential PMN defence activities, including chemotaxis, phagocytosis, respiratory burst, migration, adherence, degranulation, and apoptotic mechanisms [81,82,83]. PMN chemotaxis is modulated by extracellular ATP through P2Y2 receptors, whereas binding to P2Y receptors facilitates PMN attachment to active endothelium [81]. We here show that the ATP purinergic receptor P2X1 plays a pivotal role in *C. parvum*-induced suicidal NETosis since its inhibition by NF449 considerably blocks this defence mechanism. This conclusion is consistent with prior studies on NETosis caused by apicomplexans, such as *N. caninum* and *B. besnoiti* [38,44] and the kinetoplastid *T. b. brucei* [40], thereby indicating a general purinergic ATP receptor-mediated key mechanism in parasite-triggered NET release.

We studied for the first time the probable involvement of Notch signaling in parasite-induced NETosis in this work. This highly conserved pathway is found in almost every invertebrate to mammalian species, and is critically involved in cellular developmental processes such as homeostasis, the apoptotic process, as well as proliferation. B- and T-cell maturation in central lymphoid and peripheral tissues is regulated by Notch receptors. In spite of this, little is known about its involvement in the host’s innate immune system and its ability to regulate myeloid lineage formation and activity, especially in the setting of acute and chronic inflammation [52,84]. Owing to their constitutive expression on their cell surfaces, monocytes, macrophages, and DCs all possess the ability to activate Notch signals through TLR modulation [52,53], and it has been confirmed that activation of these cells by TLR ligands results in upregulation of Notch receptors and ligands such as DDL1/4 and Jagged [52,53,84,85,86,87,88]. Conversely, nothing is understood about the function of Notch-dependent pathways regarding PMN undergoing NETosis. Current data indicates that the Notch signaling pathway is indeed important for this defence process since respective inhibitor treatments led to diminished *C. parvum*-induced NET formation under both oxygen conditions, but barely significant. So far, it remains unclear if PMN TLRs are also involved in *C. parvum* recognition, thereby leading to Notch receptor and ligand expression. Worth noting, the pivotal roles of TLR2 and TLR4 have recently been described in bovine PMN against the closely related apicomplexan *E. bovis* [32]. Hence, *E. bovis* enhanced the expression of TLR2 and TLR4 on triggered bovine PMN and caused NF-kB activation through TLR2/4, culminating in NETosis [32] but under hyperoxia (21% O_2_). Thus, future studies under physioxia will help to elucidate the precise role of TLRs, other internal PRRs, metabolic signatures, and Notch-dependent signaling in *C. parvum*-driven bovine as well as human NETosis [2], thereby underlying the importance of experimental settings in more realistic scenarios.

## 5. Conclusions

Limitations of this study included the low number of blood donors as well as the exclusive use of one *C. parvum* strain. Furthermore, another limitation is the fact that all experiments were conducted in vitro, thereby not reflecting the in vivo intestinal scenario. Results of this study confirmed that *C. parvum* oocysts and sporozoites are significant triggers of bovine suicidal NETosis. Regarding the microaerobic *C. parvum* life cycle, which involves IEC multiplication and PMN recruitment throughout cryptosporidiosis in vivo [18,20,21] more realistic studies under physioxia are needed. Since lysis of an infected IEC exposes parasitic stages to luminal PMN, parasite trapping through NET formation may be critical. The benefits of live cell 3D holotomographic imaging for examining NETotic processes, PMN responses, and changes in response to *C. parvum* were clearly explored. The same holds true for Seahorse^®^-based glycolytic measurements, which helped to better understand complex metabolic signatures of PMN undergoing NET formation. Nonetheless, the precise function of NET formation in vivo and its role in the outcome of bovine and human cryptosporidiosis is currently unclear but it can be anticipated that this ancient and well-conserved defense mechanism will be clarified in a not too distant future.

## Figures and Tables

**Figure 1 biology-11-00442-f001:**
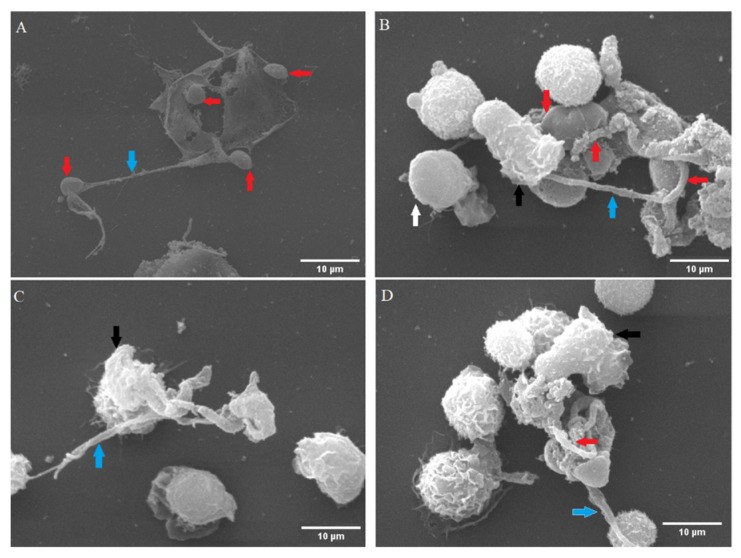
Suicidal NETosis caused by *Cryptosporidium parvum* was examined using scanning electron microscopy (SEM). Oocysts and sporozoites of *C. parvum* were fixed in the presence of fixans in the co-culture of isolated bovine PMN. NET-like filaroid structures (delicate and thick) were unveiled, as shown by the blue arrows (**A**–**D**). The presence of four firmly entrapped oocysts (red arrows) by one ruptured PMN are seen, and two captured sporozoites (marked by red arrows) are seen in extruded NETs (**B**). An almost completely covered oocyst by NETs structures (indicated by a red arrow) is also seen (**D**). Moreover, inactivated (white arrowhead, (**B**)) as well as activated PMN with rough surfaces (black arrow heads) were detected in these reactions (**B**–**D**).

**Figure 2 biology-11-00442-f002:**
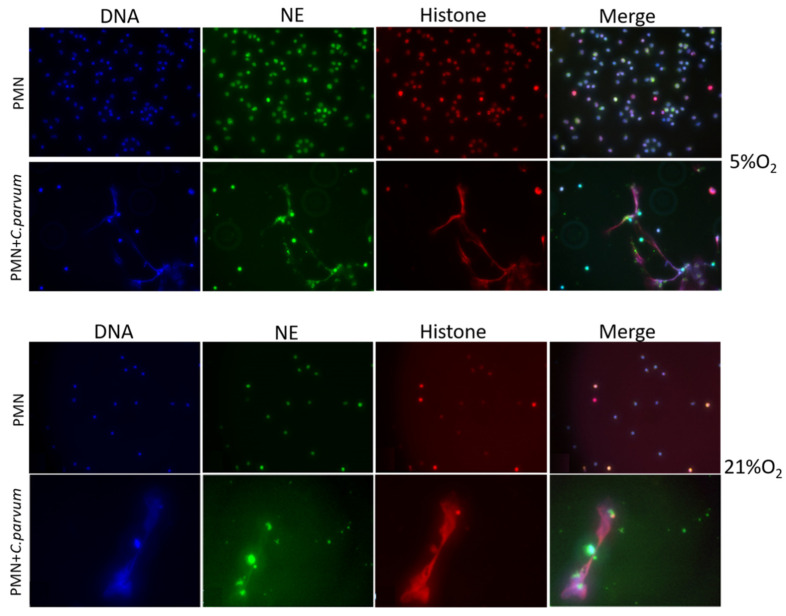
Suicidal NETosis caused by *Cryptosporidium parvum* is characterized by the co-localization of PMN-derived DNA (blue) adorned with global histones (H1, H2A/B, H3, H4, red) and neutrophil elastase (NE, green) and the resulting merging of the three channels.

**Figure 3 biology-11-00442-f003:**
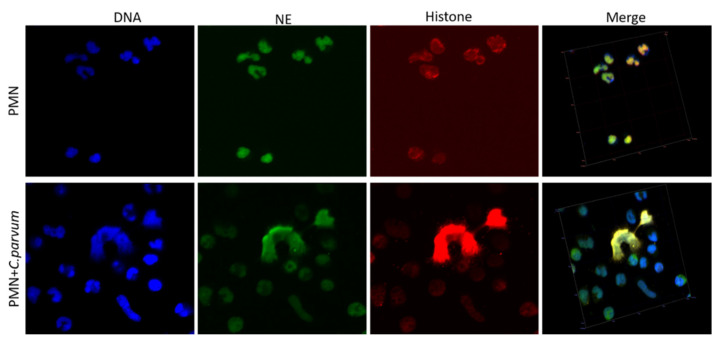
Confocal microscopy analysis demonstrates co-localization of unstimulated PMN with polymorphic nuclei (red) and granules containing neutrophil elastase (NE; green).

**Figure 4 biology-11-00442-f004:**
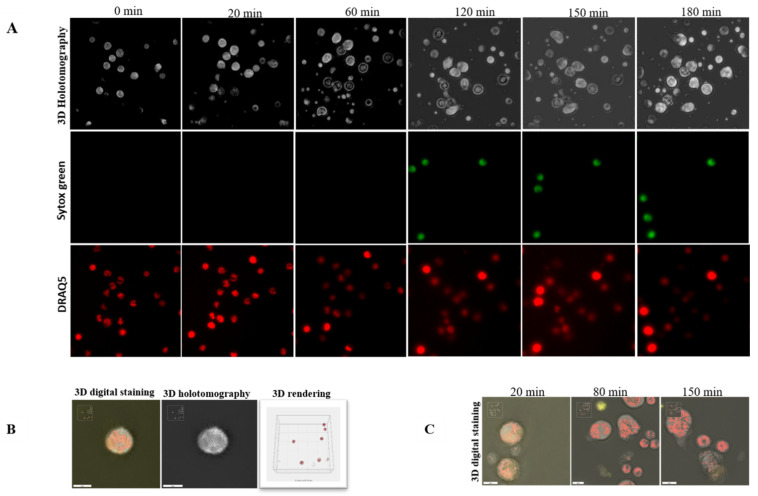
Live-cell 3D-holotomographic microscopy of PMN co-incubated with *C. parvum* (sporozoites/oocysts) in three dimensions. (**A**) is a time-lapse image of PMN stained with DRAQ5 (red) and SYTOX Green (green). (**B**) illustrates the 3D rendering, 3D holotomography picture, and 3D digital staining, of control experiment (inactivated PMN). Determination of digital staining using the refractive index (RI) values of the cell structures that were examined. The segmented nuclei region is shown in red. (**C**) Three-dimensional digital staining demonstrates the morphological alterations that parasite-exposed PMN undergo after 20, 80, and 150 min, correspondingly.

**Figure 5 biology-11-00442-f005:**
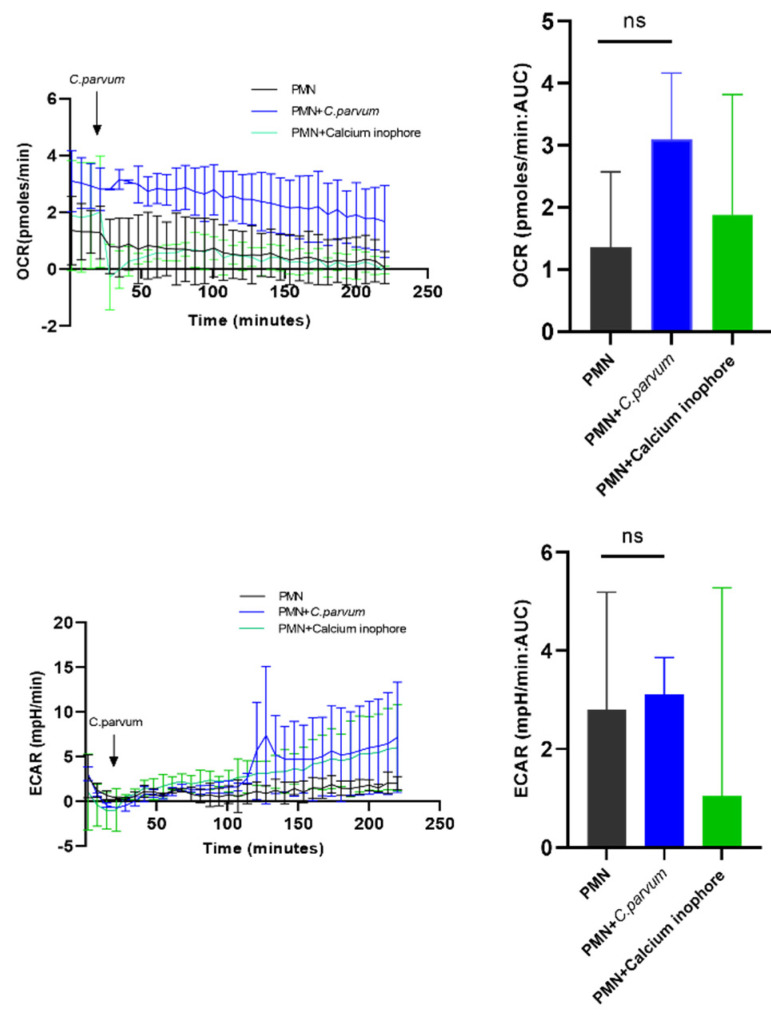
Evaluation of metabolic alterations in bovine PMN exposure to *Cryptosporidium parvum*. An extracellular flux analyzer (Seahorse^®^; Agilent) was used to measure the rate of oxygen consumption (OCR) and extracellular acidification (ECAR) in bovine PMN. For 220 min, the OCR and ECAR concentrations were measured. The area under the curve (AUC) for all registries was computed and shown as mean ± SD. Non-significant (ns).

**Figure 6 biology-11-00442-f006:**
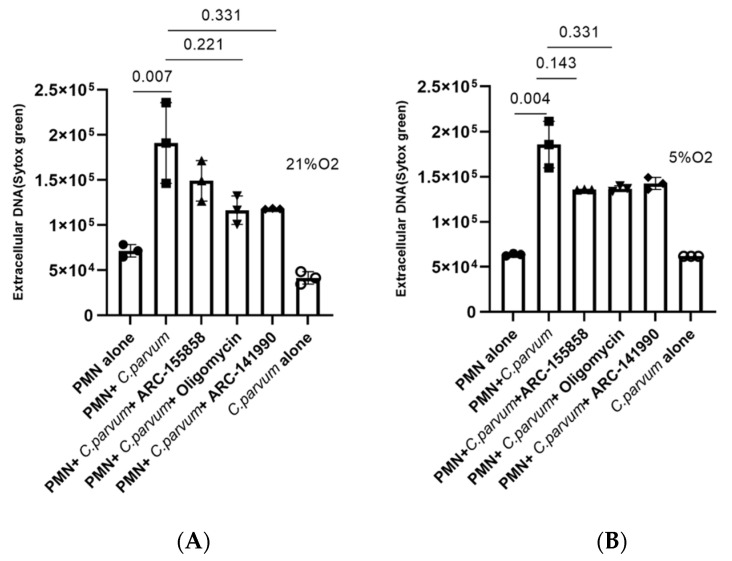
Effects of AR-C 141990-, AR-C 155858-, and oligomycin A treatments on *Cryptosporidium parvum*-triggered suicidal NETosis. Hyperoxic (**A**) and physioxic (**B**). Bovine PMN (*n* = 3) were pre-treated for 30 min with oligomycin A (5 μM), AR-C 155858 (1 μM), and AR-C 141990 (1 μM); thereafter, the cells were exposed to *C. parvum* for 3 h. Extracellular DNA was evaluated employing SYTOX Green^®^-derived fluorescence intensities measured on an automated multi-plate reader (Varioskan^®^, Thermo Scientific). In the graphs, results are reported as mean standard error of the mean, and *p* values less than 0.05 were considered statistically significant.

**Figure 7 biology-11-00442-f007:**
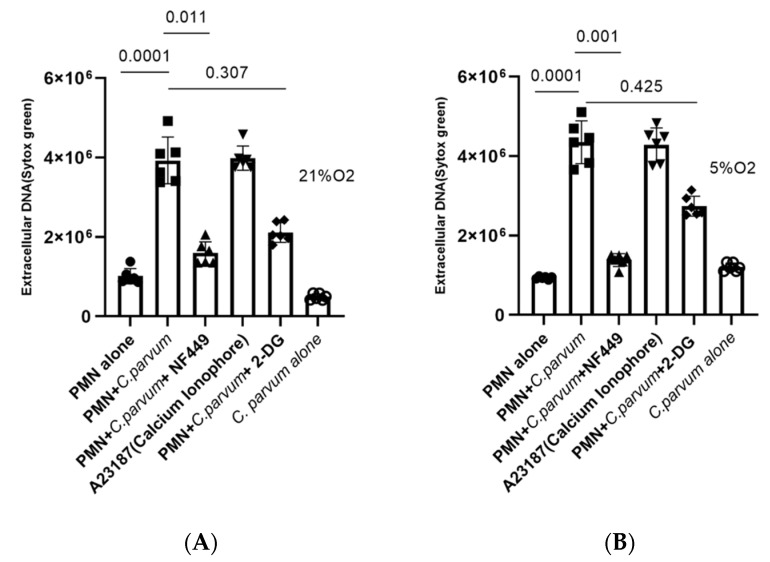
Hyperoxic (**A**) and physioxic (**B**) effects of purinergic signaling and glycolysis in *Cryptosporidium parvum*-induced NETosis (**B**). Pre-treatment of NF449 (100 μM, P2X1 inhibitor), 2 mM for 2-DG (inhibitor of 2-DG—glycolysis), were administered to bovine PMN (*n* = 6) for 30 min before exposure to *C. parvum* (preportion 1:3) and A23187 for 3 h. Extracellular DNA was evaluated employing SYTOX Green^®^-derived fluorescence intensities measured on an automated multi-plate reader (Varioskan^®^, Thermo Scientific). In the graphs, results are reported as mean standard error of the mean, and *p* values less than 0.05 were deemed statistical significant.

**Figure 8 biology-11-00442-f008:**
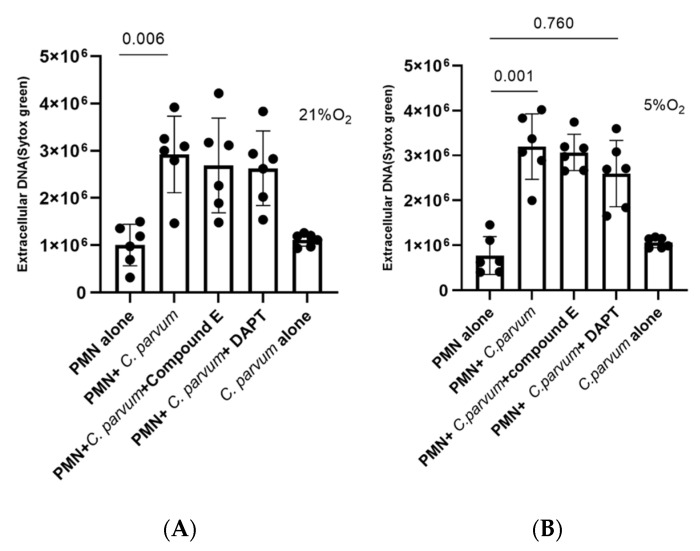
Effect of Notch inhibitors on *Cryptosporidium parvum*-induced suicidal NETs formation under hyperoxia (21% O_2_; (**A**)) and physioxia (5% O_2_; (**B**)). Pre-treatment of 50 μM for DAPT (inhibitor of Notch signaling,) and 20 nM for compound E (inhibitor of Notch signaling) were administered to bovine PMN (*n* = 6) for 30 min before exposure to *C. parvum* (preportion 1:3). Extracellular DNA was evaluated employing SYTOX Green^®^-derived fluorescence intensities measured on an automated multi-plate reader (Varioskan^®^, Thermo Scientific). In the graphs, results are reported as mean standard error of the mean, and *p* values less than 0.05 were considered statistical significant.

## Data Availability

All data generated during this study are included this article and the Appendix A.

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
