# Peer review of "ATP Purinergic Receptor P2X1-Dependent Suicidal NETosis Induced by Cryptosporidium parvum under Physioxia Conditions"

_biology, 2022, doi:10.3390/biology11030442_

Round 1

Reviewer 1 Report

Cryptosporidium parvum is a major threat to neonatal calves causing massive diarrhea and marked mucosal inflammation. Resistance and immunity to C. parvum involve host innate immune responses mediated by cells at the epithelial and subepithelial level, and acquired immune responses mediated by B and T lymphocytes. Following C. parvum infection, intestinal epithelial cells increase the expression and production of the potent neutrophil chemoattractants IL-8 and GROα. Further, infection elicites an inflammatory neutrophil infiltrate underlying the epithelium. The most notable aspect of the histopathology is infiltration of the lamina propria with predominately polymorphonuclear cells and macrophages. Numerous studies have characterised major components of adaptive immunity that are essential for control and elimination of cryptosporidial infection in humans and animals but few data are available on innate immune response and early onset response from PMN ant the present study sounds very original and enrich the knowledge of the pathophysiology of Cryptosporidium infetion. I have examined « ATP purinergic receptor P2X1-dependent suicidal NETosis induced by Cryptosporidium parvum under physioxia conditions » and find it to be a well-conducted study well documenting the in vitro activity of PMN against C. parvum oocysts and sporozoites. Overall, I found this manuscript very complete and well-written so, in my opinion, this paper could be accepted for publication. I have only some minor comments that should be considered.

In the abstract

1/ L28 “… and neonatal (is) caused…” : delete “is”

2/ L31 : “… suicidal NETosis (NETosis) in exposed bovine…” : delete “NETosis”

In the introduction

3/ L87 : “Thus PMN can detect PAMP which including microbial membrane…” : replace “including” by “include”

4/ L107-111 : The sentence  is not in accordance with reference [2]. In fact, reference 2 is not about C. parvum but only Eimeria bovis. Absolutely nothing is sais about C. parvum in this paper [2] !

In the discussion

5/ L476-477”… Bovine small intestinal epithelial cells (BSIEC) infected with C. parvum … : “infected by C. parvum is redundant in this sentence.

6/ L533 : understood (not “undrestood”)

Author Response

In the abstract

1/ L28 “… and neonatal (is) caused…” : delete “is”

“is” was deleted

2/ L31 : “… suicidal NETosis (NETosis) in exposed bovine…” : delete “NETosis”

“NETosis” was deleted

In the introduction

3/ L87 : “Thus PMN can detect PAMP which including microbial membrane…” : replace “including” by “include”

We have replaced ‘including’ for “include” in the sentence.

4/ L107-111 : The sentence  is not in accordance with reference [2]. In fact, reference 2 is not about C. parvum but only Eimeria bovis. Absolutely nothing is sais about C. parvum in this paper [2] !

We fully agree with the referee and the correct reference is now cited in the text.

In the discussion

5/ L476-477”… Bovine small intestinal epithelial cells (BSIEC) infected with C. parvum … : “infected by C. parvum is redundant in this sentence.

The sentence is now edited.

6/ L533 : understood (not “undrestood”)

Edited.

Reviewer 2 Report

This is an excellent manuscript, need only some minor revisions, as highlighted below:

Line 58: Alveolata(subphylum Apicomplexa) – space delimitation

Line 59: „neglected enteropathogen” – I not agree this idea, the cryptosporidiosis is a communicable disease within the EU countries

Line 61: „Study” – with lower case

Line 64: „properly. [8–12].” – please delete de dot after properly

Line 99: „(e.g. Eimeria bovis..” instead of „(Eimeria bovis”

Line 101: „(e.g. Haemonchus” instead of „(Haemonchus”

Line 103: „(e.g. Schistosoma” instead of „(Schistosoma”

Line 147: please complete the citation with the following representative reference: doi: 10.2807/1560-7917.ES.2018.23.4.16-00825

Line 148: „described in [4,56],” – not scientifically sound, rephrase it

Line 150, 158, 160, 167, 173, 180.... Please uniformly state manufacturer, city and country from where equipment or reagents has been sourced according to the journal style requirements. Please be carefully with this concern throughout the manuscript

Line 234: „microscope(Philips XL30®)” – please insert a space delimitation

Line 238: „co-cultured(37 °C” – please insert a space delimitation

Line 239: “verslips(15mm...” – please insert a space delimitation

Line 246: „X100(Thermo” – please insert a space delimitation

Line 266: „C. parvum(1.5 × 106)” - please insert a space delimitation

Line 268: „Green®(green channel)” – please insert a space delimitation....Revise this concern carefully throughout the manuscript.

Line 297: de-italicize the Figure 1

Lines 321-336: the format of these lines are not in agreement with the journal standards

Lines 17, 30, 35, 282, 284, 290, 293, 307, 309, 313, 322, 336, 338, 347, 355, 358, 360, 368, 376, 384, 388, 390, 397, 400, 401, 402, 404, 407, 409, 424, 425, 428, 430, 441: C. parvum – italics (please be carefully with this concern throughout the manuscript)

Line 547: I suggest to the authors to highlight the study limitations

Line 552: “we also discussed” – within this chapter, the authors needs to avoid the using of personal tense

Lines 595-596, 610, 614, 632, 635, 638, 641, 645, 717, 731, 733, 736, 741, : “parvum” with lowercase (please be carefully with this concern throughout the reference list)

Author Response

Line 58: Alveolata (subphylum Apicomplexa) – space delimitation

The space  was inserted.

Line 59: „neglected enteropathogen” – I not agree this idea, the cryptosporidiosis is a communicable disease within the EU countries

We agree with you, therefore the sentence was edited and the word ‘’neglected’’ was remove and replaced by ‘’important’’.

Line 61: „Study” – with lower case

Changed to “study”

Line 64: „properly. [8–12].” – please delete de dot after properly

The dot is now deleted.

Line 99: „(e.g. Eimeria bovis..” instead of „(Eimeria bovis”

We did so as you mentioned.

Line 101: „(e.g. Haemonchus” instead of „(Haemonchus”

We did so as you mentioned.

Line 103: „(e.g. Schistosoma” instead of „(Schistosoma”

We did so as you mentioned.

Line 147: please complete the citation with the following representative reference: doi: 10.2807/1560-7917.ES.2018.23.4.16-00825

The reference was cited in the text.

Line 148: „described in [4,56],” – not scientifically sound, rephrase it

This sentence was rephrased with a better word (as outlined previously).

Line 150, 158, 160, 167, 173, 180.... Please uniformly state manufacturer, city and country from where equipment or reagents has been sourced according to the journal style requirements. Please be carefully with this concern throughout the manuscript

We put all information according to your comment throughout the manuscript.

Line 234: „microscope(Philips XL30®)” – please insert a space delimitation

The space was inserted.

Line 238: „co-cultured(37 °C” – please insert a space delimitation

The space was inserted.

Line 239: “verslips(15mm...” – please insert a space delimitation

The space was inserted.

Line 246: „X100(Thermo” – please insert a space delimitation

The space was inserted.

Line 266: „C. parvum(1.5 × 106)” - please insert a space delimitation

The space was inserted.

Line 268: „Green®(green channel)” – please insert a space delimitation....Revise this concern carefully throughout the manuscript.

The space was inserted and kept it throughout revised manuscript.

Line 297: de-italicize the Figure 1

Corrected.

Lines 321-336: the format of these lines are not in agreement with the journal standards

We changed the lines format to the journal standard.

Lines 17, 30, 35, 282, 284, 290, 293, 307, 309, 313, 322, 336, 338, 347, 355, 358, 360, 368, 376, 384, 388, 390, 397, 400, 401, 402, 404, 407, 409, 424, 425, 428, 430, 441: C. parvum – italics (please be carefully with this concern throughout the manuscript)

All changed to italics.

Line 547: I suggest to the authors to highlight the study limitations

We once again agree with this comment, and therefore we included the following sentence in revised manuscript:

“Limitations of this study included the low number of blood donors as well as the use of exclusively one C. parvum strain. Furthermore, another limitation of this study is the fact that all experiments were conducted in vitro thereby not reflecting in vivo intestinal scenario”. 

Line 552: “we also discussed” – within this chapter, the authors needs to avoid the using of personal tense

The sentence was edited.

Lines 595-596, 610, 614, 632, 635, 638, 641, 645, 717, 731, 733, 736, 741, : “parvum” with lowercase (please be carefully with this concern throughout the reference list)

We corrected all Parvum to parvum.